# Stadium attendance demand in the men's UEFA Champions League: Do fans value sporting contest or match quality?

**George Wills[1], Francesco Addesa[2]\*, Richard Tacon[1]**

**1** Department of Management, Birkbeck, University of London, London, United Kingdom, **2** Sport Management Group, Carnegie School of Sport, Leeds Beckett University, Leeds, United Kingdom

\* f.a.addesa@leedsbeckett.ac.uk

**Data Availability Statement:** The dataset used is available from the Figshare database (DOI: 10.6084/m9.figshare.20979511.v1).

## Abstract

This paper is the first to empirically analyse the determinants of stadium attendance demand in the men's UEFA Champions League, the most prestigious competition in club football. The analysis covers 1,234 matches from 2009/10 to 2018/19 across 32 nations. The results show that outcome uncertainty and competitive intensity are not significantly associated with higher attendances, but the level of team quality is, for all fans, and the presence of star players is, for fans of clubs outside the top five European leagues. The empirical analysis—based on Tobit model regressions—enables an evidence-informed discussion of the competition structure of the UCL and the highly charged debate surrounding a potential European Super League. The article also offers insights for the wider body of academic knowledge on stadium attendance demand, by adding rare analysis of an international cup competition and an improved understanding of the connection between star players and fan interest in European football.

## Introduction

Understanding the determinants of stadium attendance is vital from a behavioural economics perspective, enhancing our understanding of why people choose to watch live sport and enabling sport organisations to structure events in ways that maximise utility for consumers. The popular claim that competitions need to be evenly balanced in order to be attractive to fans, based upon Simon Rottenberg's [1] uncertainty of outcome hypothesis (UOH), underpins the policy choices of many sport organisations. Indeed, the current 5-year strategy of UEFA, the governing body of football in Europe, outlines competitive balance as a priority for the organisation and commits to 'develop and implement specific regulations aimed at preserving and improving competitive balance' [2]. In contrast, the majority of academic studies analysing the impact of competitive balance on stadium attendance report the opposite effect, finding that attendance is maximised when either the home or away team has a significantly higher change of winning [3]. For this reason, the question about the importance of uncertainty of outcome is a live one and, given how much it affects the policy of sport organisations, the structure of sporting competitions and, ultimately, sport fans, athletes, broadcasters and sponsors, it remains crucial.

**Funding:** The author(s) received no specific funding for this work.

**Competing interests:** The authors have declared that no competing interests exist.

Since the 1950s, a great deal of academic research has focused on better understanding the determinants of stadium attendance through empirical analysis. Indeed, a landmark scoping review found 235 analyses of stadium attendance demand for sport worldwide [4]. Around half of these focused on European football, making this by far the most studied field. However, as Schreyer and Ansari [4] point out, the vast majority of these studies have focused on individual domestic leagues, meaning there is a lack of understanding of both international football competitions and cup format tournaments. In particular, to this point, there have been no studies that analyse the determinants of stadium attendance in the men's UEFA Champions League (UCL). This is surprising, because the UCL is generally considered the most prestigious tournament in club football. Moreover, the determinants of fan interest in the UCL have been hotly disputed since the 1980s, as part of a wider debate over the potential for a breakaway European Super League (ESL), i.e., the proposed formation of an additional tier of European football outside the traditional pyramid format of UEFA and the national football associations. This is particularly salient right now, as in April 2021, 12 elite European football clubs attempted to break away from the UCL to form a European Super League, fuelled in part by arguments about what fans want to watch [5]. At the same time, UEFA, the competition organiser, announced significant changes to the UCL competition format, which had stood largely unchanged for over 20 years, to modernise the competition and prevent a potential breakaway league. This paper offers the first empirical study of the determinants of stadium attendance in the men's UCL across a full decade of matches from 2009/10 to 2018/19 within 32 different nations. As such, we hope it will make a major contribution to the academic literature on stadium attendance demand and will also offer empirical insight into the ongoing debate about how best to organise the UCL and European club football more broadly.

This study also offers a novel approach to the analysis of stadium attendance demand in European football, by considering the effect of both overall team quality *and* individual star players. While the use of individual star players as a measure of match quality is well established in North American stadium attendance demand literature [6–12], the dominant measure in European football has been aggregate team quality. Part of the reason for this may be that European football stars tend to be split across different national leagues [13]. In this sense, the UCL–the only competition in club football where the world's best players and teams compete every year–offers the rare opportunity to consider the impact of both team quality and individual star players on stadium attendance demand.

Overall, this study looks to bring new understanding to the debate over the validity of the outcome uncertainty hypothesis in European football and further insight into the determinants of stadium attendance in the UCL, in European football and in sport more broadly. It will increase the breadth of understanding in European football through the inclusion of individual star players as a variable of demand, give a rare insight into fan demand for cup competitions and consider fan behaviour across 32 separate nations.

The article is structured as follows. The next section reviews the academic literature on stadium attendance demand in sport, examining previous work on the impact of the sporting contest (uncertainty of outcome and competitive intensity) and match quality (team quality and star players). The subsequent sections set out the data and methods used in the analysis, present the key findings, before moving to a broader discussion of the implications of these findings for the UCL and the threat posed by a breakaway European Super League.

## Stadium attendance demand literature

Simon Rottenberg's [1] analysis of baseball players' labour markets offered a seminal insight into the economics of stadium attendance. Rottenberg's study proposed that attendance was a

function of income levels, population size, team quality, the availability of substitutes and, notably, the competitive balance of the match. This early analysis of competitive balance formed the uncertainty of outcome hypothesis, which states that closer sporting contests lead to great spectator interest and larger stadium attendances. This hypothesis was reinforced by Walter Neale's [14] analysis of the 'peculiar economics' of sports leagues, where the greater the economic collusion and the more the sporting competition, the greater the profits. For Neale, the most important determinant of stadium attendance was match quality, which increased the demand for game admissions and in turn drove demand for skilled players. Roger Noll [15] expanded knowledge in this area, studying the determinants of demand in the four largest North American sports, concluding that attendance was higher in larger cities and that the quality of the team and players attracted more fans to matches.

These early works built a strong foundation for understanding stadium attendance demand, although as Noll [16] later pointed out, 'the original papers do not contain a single regression'. The following 50 years saw a proliferation of studies to better understand stadium attendance demand–and these were certainly not short of regressions. Indeed, in the landmark review noted above, Schreyer and Ansari [4] undertook the most comprehensive analysis of empirical research on stadium attendance demand and included 235 regression-based empirical studies in 195 manuscripts across 13 different sports, up to August 2020. By far the most (107) focused on European football, followed by the four largest North American sports, although sports such as NASCAR [17], Ultimate Fighting [18], handball [19] and tennis [20] have also been analysed. Within this scoping review, Schreyer and Ansari [4] found the dominant manuscript themes to be the role of sporting contest characteristics, especially outcome uncertainty, and the role of star players and team success. These key themes form the focus of our paper and we discuss them in more detail below. Schreyer and Ansari's [4] review also highlighted specific gaps in knowledge in this area and we return to these at the end of this section.

## Sporting contest (outcome uncertainty and competitive intensity)

The uncertainty of outcome hypothesis (UOH) underpins many of the policies within professional sports markets and, as a result, the importance of outcome uncertainty to stadium attendance demand has become a key area of debate [21–31]. Indeed, Schreyer and Ansari's [4] review found that the titles of roughly a quarter of all the studies they included referred to the concept of outcome uncertainty.

However, the findings from empirical research on the importance of outcome uncertainty are not conclusive [32–36]. Within football specifically, a number of studies across both American and European leagues [21,23,36–43] have found that there is a limited relationship between outcome uncertainty and stadium attendance. As a result, some researchers have proposed an alternative view that the relationship between outcome uncertainty and stadium attendance follows a reference-dependent preferences model [44,45]. Based on Prospect Theory [46], which proposes that individuals assess their loss and gain perspectives in an asymmetric manner, it has been suggested that fans' decisions to attend are influenced by the opportunity to watch their team either play a far inferior opponent (loss aversion) or cause an upset result against a stronger team (the 'David and Goliath effect'). In this model, fan demand instead follows a U-shaped curve, with demand increasing as match certainty increases in either direction. Despite this debate surrounding the validity of the UOH, the theoretical assumption that fans prefer more balanced contests continues to underpin policies in North American and European sports aimed at redistributing resources, including salary caps, reverse-order drafts and revenue sharing.

Beyond uncertainty of outcome, an additional measure of sporting contest, first proposed by Kringstad and Gerrard [47,48], is competitive intensity. While uncertainty of outcome represents the importance of relative balance between teams, competitive intensity is based on the uncertainty of outcome in relation to sporting stakes. The assumption for this measure is that, alongside uncertainty of outcome, fans are also motivated by the level of sporting prize that is at stake during the match. Scelles et al. [42] were the first to empirically analyse the effect of competitive intensity on stadium attendance demand and numerous studies have since incorporated it [21,23,42,49–51]. These studies have found that competitive intensity has a significant impact on consumer interest across different European football competitions, albeit it to differing degrees dependent upon the competition and sporting prize in question. Our study will examine the effect of both uncertainty of outcome and the level of competitive intensity on stadium attendance demand.

## Match quality (team quality and star players)

Characteristics demand theory [52] proposes that all goods possess characteristics that are demanded by consumers. For stadium attendees of sporting events, these characteristics can be defined as the overall match quality on show. Researchers have previously captured this demand through variables for both overall team quality and the presence of star players. Within European football, team quality has been the dominant measure of match quality. It has been operationalised in a variety of ways, including home and away team budgets, team reputation, the current and past season winning percentages, the expected quality of teams at the start of the season and the number of goals a team has scored [21,23,35,36,41–43,53–55]. Serrano et al. [31] analysed match attendance in four major European football leagues during the 2012/13 season and found that the most important variable was team quality, measured by the value of the players taking part. Further studies of individual leagues have found team quality to be a significant factor for stadium attendances in the English Premier League [24,37], Spanish La Liga [55], Italian Serie A [23] and French Ligue 1 [42].

The 'star player effect' was established by Rosen's model [56], which predicted that a small number of athletes with marginally more talent than their peers will attract far more fans who will pay to watch them. Academic literature on the impact of individual star players on stadium attendance is very well developed in the North American sports markets, with researchers finding a positive association between the two in the NBA [6,7,9,57,58], Major League Baseball [11,12,59] and the NFL [60]. Star players' impact on stadium attendance has also been captured in other sports, including tennis [20], boxing [61], cricket [62], the Japanese Professional Baseball League [63] and the Ultimate Fighting Championship [64]. Within football, analysis of the effect of individual star players on stadium attendance demand has mainly focused on Major League Soccer [8,10,39,43,65,66], with top earning players, MLS All-Star players and designated players all found to attract higher stadium attendances. Studies have also found a positive effect of star players on attendance in the U.S. Women's Professional Soccer League [67] and the Chinese Super League [68].

In European football, however, there has been almost no empirical research directly examining the effect of individual star players. To date, the only study that examines the 'superstar effect' on stadium attendance is by Brandes et al. [69], who found that, within the German Bundesliga, 'local heroes' enhanced home match attendance and 'star players' increased attendance both at home and on the road. Meanwhile, Wills et al. [13] examined the effect of star players on TV audiences for the UCL and found a significant, positive effect. This lack of research is surprising, especially given the wealth of literature on stadium attendance demand in European football [4] and the findings from the North American research, i.e., that star

players often have an impact on fan interest over and above the impact they have on the ability or success of the team they play for. This lack of focus on individual star players may be, in part, because European football stars tend to be split across different national leagues [13]. In this sense, the UCL–the only competition in club football where the world's best players and teams compete every year–offers the rare opportunity to consider individual star players as a variable within European football. Our study will seek to expand understanding in European football by examining the effect of both team quality *and* the presence of star players on stadium attendance demand.

## New stadium attendance demand insights–UEFA Champions League

Schreyer and Ansari's [4] scoping review of stadium attendance demand research found that 'half of all the studies centred on stadium attendance demand for European football' (p.14) and their measure of study frequency by sport predicts that the gap between European football and other sports is likely to widen in the future. Yet despite this focus on European football, Schreyer and Ansari [4] identified notable gaps in stadium attendance demand analysis, in particular the lack of studies on 'domestic and international cup competitions' (p.23). Across the scoping review, just four studies focused on international tournaments [51,70–72] and Schreyer and Ansari [4] name-checked the men's UEFA Champions League as a competition that ought to be examined, with only one published study to date on the women's version [51]. Wills et al. [13] analysed the determinants of TV demand in the men's UCL, but the scope of this analysis was limited to six nations, due to restrictions on the data available. This lack of understanding of the men's UCL is particularly surprising, as it is widely regarded as the most prestigious competition in club football and is the only competition where the elite clubs and players from all of Europe's top-tier football leagues can meet. This paper will look to fill this gap, by offering the first empirical study of the determinants of stadium attendance in the men's UCL. Due to the greater availability of stadium attendance data, this study is also able to consider the demands of consumers in all 32 nations that took part in the UCL from 2009/10 to 2018/19. This allows us to separately analyse and compare the determinants of stadium attendance for clubs in the top five leagues, which are largely those that have the potential to join a European Super League, and the clubs outside those leagues.

## Methodology

Our analysis focuses on 1,234 UCL matches played from 2009/10 to 2018/19. Over this time, there has been no change to the format of the competition: the winners and runners-up from each of the eight groups in the group stage qualify for the knockout stage, which comprises a round of 16, quarter-finals, semi-finals and the final. 16 matches were omitted from the data-set: all the finals, as they are played in neutral stadiums, and six group-stage matches where the home team was serving a stadium ban.

The demand model to be estimated is based on Valenti et al.'s [51] study on the determinants of stadium attendance in the women's UCL:

$$ln(attendance_{ijt}) = \alpha X_{ijt} + \beta Z + \gamma S + \delta T + e_{ijt}. \tag{1}$$

Our dependent variable is the overall number of spectators for each match–obtained from individual match data on skysports.com, whereas $X_{ijt}$ is a vector of independent variables, $Z$ a vector of dummy variables, $S$ a vector of dummies capturing season fixed effects, $T$ a vector of dummies capturing team fixed effects, $\alpha$, $\beta$, and $\gamma$ the associated coefficients, and $e_{ijt}$ the disturbance term. The use of the overall number of spectators as dependent variable implies some limitations, as it does not account for spectator no-show behaviour [73–75], that could be

particularly relevant during the group stages, and the number of free tickets, and does not differentiate among season and match-day tickets. Nonetheless, it is still the dependant variable most commonly employed in studies on stadium demand [4].

As pointed out by Borland and MacDonald [76], demand for live sport is constrained by supply capacity, represented in this case by the number of seats available in a stadium. Therefore, to account for the truncation of the number of spectators at the upper boundary, the estimation method is a Tobit regression model with individual cut-off points [77], where the stadium capacity is used as an individual cut-off point. 29 observations within the Tobit model were then right censored for the whole dataset. Regressions were performed using the metobit command in Stata 17 software [78].

The explanatory variables are aligned to Valenti et al. [51] and presented according to Borland and MacDonald's [76] categorisation of the determinants of demand for sport. Among the economic factors potentially impacting on stadium attendance, we include the income per capita of the home and away club's region (*home_income* and *away_income* respectively) as an indicator of the fans' purchasing power. This is measured as the Gross Domestic Product at current market prices in the NUTS 3 region of each club, as defined by the European Commission on ec.europa.eu. The use of NUTS 3 regions is intended to capture the variations in purchasing power between different regions within countries. The distance between the home and away club's location (*distance*), a proxy for the travelling costs for away supporters, is measured as the distance in kilometres between each club's stadium location on distancefromto. net. To account for factors determining the quality of viewing, as defined by Borland and MacDonald [76], we included an integer (*temperature*) and a dummy (*rain*) variable for weather conditions, with the data from match reports on uefa.com.

The last set of variables–the most relevant to this study–relates to the sporting contest [76]. We first included two variables–*home_team_form* and *away_team_form*–capturing the total points won from each team's most recent five UCL matches, obtained from individual match data on uefa.com. There is an expected positive effect of the two teams' recent performances on stadium attendance [41]. Then, to account for the widely discussed uncertainty of outcome hypothesis (*outcome_uncertainty*), we used the absolute difference between the home and the away team win probabilities [38] rather than draw probabilities, as the former is more sensitive to the actual gap between teams [23,79]. Data on the historical betting odds for each individual match were obtained from oddsportal.com, with the average odds used from all bookmakers in each individual game. The specific choice of bookmakers is considered irrelevant as the correlations between posted odds by different bookmakers are extremely high [80]. To account for bookmaker margins the adjusted probabilities are obtained by dividing each of the probabilities (home win, away win, draw) by the sum of the unadjusted probabilities so that the adjusted values sum to 1.

Subsequently, we included variables capturing the degree of relevance of a match. Since we expect stadium attendance to increase as the competition progresses, four dummies were used to account for the stage of the competition: *group_stage*, *last_16*, *quarter_final* and *semi_final*. Another dummy variable–*derby*–was included to account for the potentially higher crowd attracted by a more intense rivalry [80,81], in this case linked to matches involving teams from the same country. Competitive intensity is measured by two different variables, previously used in Valenti et al. [51] and Wills et al. [13], due to the format of the UCL. The first variable (*competitive_intensity_ko*) aims to capture the potential for score reversals after the first leg matches in the knockout stage. This variable is equal to zero before the first leg, and to the absolute goal difference after the first leg. The 'away goal' rule applied in the UCL, i.e., that any goals scored away count double, is also used here when the first leg aggregate score is equal. We expect that a larger gap in the score between the two teams would determine a lower

attendance in the second leg. Moreover, there may be situations in the group stage–usually in the last two fixtures–where there is no qualification at stake for either team to either the UCL or Europa League knockout stage (as some teams may have already been eliminated or relegated to the Europa League knockout stage, or may have already advanced to the UCL last 16). Therefore, the second variable (*competitive_intensity_gs*) is equal to 1 where there is no qualification at stake, and 0 otherwise. Group stage matches with no qualification at stake are expected to attract fewer spectators.

Finally, we consider the variables capturing match quality. First, we assessed the quality of the teams involved in the match (*home_team_quality* and *away_team_quality*), using standardised roster values, calculated as the ratio between a team's roster value and the seasonal average roster value [79]. The roster values for each club are derived from the seasonal market valuations given to each club's squad on transfermarkt.co.uk. Then, in order to verify whether the presence of individual talented players may be appealing to fans, we included our star player variables, based on the results of the Ballon d'Or competition. This is an annual competition which involves football journalists, coaches and captains of national teams voting on the male player deemed to have performed the best over the previous year. Having started in 1956, this is generally recognised as the most prestigious individual award for players. Since 1995, it has recognised players of all nationalities active at European clubs (further expanded to all clubs worldwide from 2007) and consequently represents an ideal measure for the star players competing in the UCL [13]. The use of Ballon d'Or results is directly comparable to studies in the North American context, such as Hausman and Leonard's [81] use of 'All-Star' players in the NBA and Rivers & DeSchriver's [12] use of MVP and Cy Young Award voting in the MLB as a measure of star players. *bd_winner* is a dummy equal to 1 if the current winner of the Ballon d'Or played for one of the teams involved in the match, whereas *bd_3* and *bd_10* measure how many players voted in the top three and in the top ten of that year's Ballon d'Or were involved in the match, respectively. *bd_winner_1*, *bd_3_1* and *bd_10_1* are similar to the aforementioned variables, but account for the results of the previous year's Ballon d'Or competition.

The descriptive statistics for both dependent and explanatory variables are shown in Table 1.

## Empirical findings

The findings of our analysis are presented in Tables 2–4.

Seven different specifications of the demand model have been tested. In the first specification, we have only considered the two teams' quality, whereas in the other six the team quality variables have been excluded and the star player variables included. We did this to clearly differentiate the effects of these two types of variables, first because star players are inevitably linked with the teams with the highest roster value and second because the UCL offers a first opportunity to consider the impact of individual star players on attendances in European football. First, the regressions have been run by using the whole dataset (Table 2). Then, in order to investigate whether the determinants of stadium attendance have a different impact depending on whether fans belong to clubs in the top five European leagues (i.e., those more accustomed to attending matches with more talented clubs and players and whose clubs have the potential to join a ESL), or to clubs belonging to the other European leagues, we have split the dataset into two sub-groups: 746 matches played in the countries hosting the top five European leagues (England, France, Germany, Italy, Spain) and 488 matches played in the other countries (Table 3 and 4 respectively). All the continuous explanatory variables are expressed in natural logs to interpret the estimated coefficients as elasticities. The absence of strong

**Table 1. Descriptive statistics of dependent and explanatory variables.**

| Variable | Mean | Std. Dev. | Min | Max |
|---|---|---|---|---|
| Attendance | 44525.28 | 19884.32 | 3663 | 99000 |
| home_income | 41334.08 | 29999.86 | 2526 | 185741 |
| away_income | 41236.59 | 29982.16 | 2526 | 185741 |
| Distance | 1541.45 | 899.24 | 0 | 6163 |
| Temperature | 11.75 | 7.08 | -9 | 32 |
| Rain | 0.09 | 0.29 | 0 | 1 |
| home_team_form_ | 0.55 | 0.22 | 0 | 1 |
| away_team_form | 0.57 | 0.23 | 0 | 1 |
| outcome_uncertainty | 0.41 | 0.22 | 0 | 0.80 |
| group_stage | 0.13 | 0.33 | 0 | 1 |
| last_16 | 0.06 | 0.25 | 0 | 1 |
| quarter_final | 0.03 | 0.18 | 0 | 1 |
| Derby | 0.02 | 0.12 | 0 | 1 |
| competitive_intensity_ko | 0.18 | 0.63 | 0 | 5 |
| competitive_intensity_gs | 0.02 | 0.15 | 0 | 1 |
| home_team_quality | 41334.08 | 29999.86 | 2526 | 185741 |
| away_team_quality | 41236.59 | 29982.16 | 2526 | 185741 |
| bd_w | 0.09 | 0.29 | 0 | 1 |
| bd_3 | 0.25 | 0.59 | 0 | 3 |
| bd_10 | 0.73 | 1.23 | 0 | 7 |
| bd_w_1 | 0.10 | 0.33 | 0 | 3 |
| bd_3_1 | 0.26 | 0.58 | 0 | 3 |
| bd_10_1 | 0.73 | 1.23 | 0 | 6 |

collinearity is proved by the value of the variance inflation factors (VIF) for the independent variables, that are all significantly lower than 10. Moreover, the Breusch-Pagan test reveals the presence of heteroskedasticity: therefore, robust standard errors are estimated.

## Sporting contest (outcome uncertainty, team form and competitive intensity)

The first interesting finding is that uncertainty of outcome is only significant in one specification, for fans of clubs in the top five leagues when including team quality among the explanatory variables, and is not significant in all 20 other specifications. This is consistent with other recent research that does not find outcome uncertainty to be particularly, or at all, significant in driving attendance in European football [21,23,36,37,41,42,82]. In fact, the findings show weak evidence of a *negative* effect of outcome uncertainty on attendance for clubs in the top five leagues, whose fans may be more loss averse [44], and no effect at all for clubs in the other countries, whose fans are likely to be aware of the gap with the top UCL clubs and are instead attracted by the quality of the opponents. It is worth saying that these findings do not mean that fans are entirely unmotivated by uncertainty of outcome and that, if the outcome was known in advance, stadium attendance demand would still be high. Instead, it is to say that within the broader context of European football, where there is obviously a certain level of outcome uncertainty, tightly balanced matches do not seem to have a specific, significant effect on fans' behaviour in attending the stadium.

Although our research suggests that outcome uncertainty is not a significant factor, the results on team form do suggest that fans are still somewhat influenced by a feature that might

**Table 2. UCL attendance, all matches.**

|  | (1) | (2) | (3) | (4) | (5) | (6) | (7) |
|---|---|---|---|---|---|---|---|
| home_income | 0.522*** | 0.533*** | 0.516*** | 0.524*** | 0.539*** | 0.537*** | 0.530*** |
|  | (0.051) | (0.052) | (0.052) | (0.051) | (0.054) | (0.053) | (0.054) |
| away_income | 0.019* | 0.043*** | 0.042*** | 0.040*** | 0.042*** | 0.042*** | 0.039*** |
|  | (0.010) | (0.010) | (0.010) | (0.010) | (0.010) | (0.010) | (0.010) |
| distance | -0.002 | -0.002 | -0.004 | -0.004 | -0.001 | -0.003 | -0.005 |
|  | (0.009) | (0.009) | (0.009) | (0.009) | (0.009) | (0.009) | (0.009) |
| temperature | 0.003** | 0.003** | 0.003** | 0.003** | 0.003** | 0.003** | 0.003** |
|  | (0.001) | (0.001) | (0.001) | (0.001) | (0.001) | (0.001) | (0.001) |
| rain | 0.014 | 0.009 | 0.010 | 0.011 | 0.011 | 0.011 | 0.013 |
|  | (0.018) | (0.019) | (0.019) | (0.019) | (0.019) | (0.019) | (0.019) |
| home_team_form | 0.118 | 0.174* | 0.171* | 0.172* | 0.168* | 0.171* | 0.174* |
|  | (0.096) | (0.099) | (0.098) | (0.098) | (0.099) | (0.098) | (0.099) |
| away_team_form | -0.020 | 0.157** | 0.118 | 0.117 | 0.155* | 0.124 | 0.105 |
|  | (0.081) | (0.079) | (0.080) | (0.079) | (0.080) | (0.079) | (0.079) |
| outcome_uncertainty | 0.073 | -0.026 | -0.029 | -0.026 | -0.025 | -0.031 | -0.022 |
|  | (0.049) | (0.044) | (0.044) | (0.044) | (0.044) | (0.044) | (0.044) |
| group_stage | -0.055** | -0.090*** | -0.080*** | -0.081*** | -0.089*** | -0.074*** | -0.069** |
|  | (0.024) | (0.025) | (0.026) | (0.026) | (0.024) | (0.025) | (0.025) |
| last_16 | 0.027 | 0.014 | 0.023 | 0.020 | 0.014 | 0.026 | 0.027 |
|  | (0.025) | (0.026) | (0.027) | (0.026) | (0.025) | (0.026) | (0.026) |
| quarter_final | 0.005 | 0.005 | 0.011 | 0.008 | 0.004 | 0.011 | 0.007 |
|  | (0.027) | (0.027) | (0.028) | (0.028) | (0.027) | (0.027) | (0.027) |
| derby | 0.014 | 0.013 | 0.003 | -0.001 | 0.024 | 0.017 | 0.018 |
|  | (0.040) | (0.039) | (0.039) | (0.039) | (0.039) | (0.040) | (0.040) |
| competitive_intensity_ko | -0.004 | -0.006 | -0.006 | -0.006 | -0.006 | -0.005 | -0.005 |
|  | (0.008) | (0.008) | (0.008) | (0.009) | (0.008) | (0.008) | (0.009) |
| competitive_intensity_gs | -0.065 | -0.058 | -0.059 | -0.059 | -0.063 | -0.066 | -0.070 |
|  | (0.053) | (0.052) | (0.052) | (0.052) | (0.052) | (0.052) | (0.052) |
| home_team_quality | 0.004 |  |  |  |  |  |  |
|  | (0.142) |  |  |  |  |  |  |
| away_team_quality | 0.248*** |  |  |  |  |  |  |
|  | (0.035) |  |  |  |  |  |  |
| bd_winner |  | 0.059** |  |  |  |  |  |
|  |  | (0.027) |  |  |  |  |  |
| bd_3 |  |  | 0.054*** |  |  |  |  |
|  |  |  | (0.015) |  |  |  |  |
| bd_10 |  |  |  | 0.026*** |  |  |  |
|  |  |  |  | (0.070) |  |  |  |
| bd_winner_1 |  |  |  |  | 0.041** |  |  |
|  |  |  |  |  | (0.019) |  |  |
| bd_3_1 |  |  |  |  |  | 0.052*** |  |
|  |  |  |  |  |  | (0.013) |  |
| bd_10_1 |  |  |  |  |  |  | 0.028*** |
|  |  |  |  |  |  |  | (0.007) |
| constant | 3.886*** | 3.675*** | 3.900*** | 3.837*** | 3.608*** | 3.653*** | 3.779*** |
|  | (0.572) | (0.605) | (0.602) | (0.594) | (0.624) | (0.610) | (0.625) |
| Sigma | 0.043*** | 0.045*** | 0.044*** | 0.044*** | 0.045*** | 0.044*** | 0.044*** |

*(Continued)*

**Table 2.** (Continued)

| | (1) | (2) | (3) | (4) | (5) | (6) | (7) |
|---|---|---|---|---|---|---|---|
| | (0.004) | (0.004) | (0.004) | (0.004) | (0.004) | (0.004) | (0.004) |
| *Observations* | 1234 | 1234 | 1234 | 1234 | 1234 | 1234 | 1234 |

*Notes*. Robust standard errors in parentheses obtained using the robust or sandwich estimator of variance: $p^* < 0.10$, $p^{**} < 0.05$, $p^{***} < 0.01$.

*semi_final* omitted because of collinearity.

Season and team fixed effects omitted for sake of simplicity.

lead to a more balanced match. These findings show that better form for the away team has a significant positive impact on attendances in the top five leagues in all but one specification, whereas better form for the home team has a significant positive impact on attendances in the other leagues in all but one specification. In both scenarios, this represents the team which has the longest odds of winning the match. In the top five leagues, a €1 bet on the home team would on average return €2 for a win, whereas the away team would on average return €6.50, showing the away team is the clear underdog in most matches. In the other leagues, a €1 bet on the home team would on average return €3 for a win, whereas the away team would on average return €2.40, this time showing that the home team is more often the underdog. Therefore, the findings suggest that fans are more attracted to a match when the weaker team is experiencing a run of better form in the competition.

Finally, there is no evidence of an impact of competitive intensity on attendance in the analysis conducted on the whole dataset as well as within the top five leagues in either the group stage or knockout rounds. There is also no impact of competitive intensity on attendance during the knockout rounds in the other countries, although there is a weak significance during the group stage. This is unusual within European football, as previous studies that included this variable found at least a weak positive effect on stadium attendances [21,23,42,50,51]. Our finding could in part be due to the fact that most tickets for the group stage are sold before the fans know whether the final matches will be critical for qualification, and tickets in the knockout stage are sold before fans know the result of the first leg. It may also be because there are more star players and higher quality teams in the UCL, meaning that the level of competitive intensity in each match is less essential than for domestic leagues.

## Match quality (team quality and star players)

Overall, the findings suggest that fans are primarily attracted by the quality of the teams and players on show. If we consider the whole dataset, only the away team quality shows a positive impact on attendance. However, some differences emerge when analysing the two different subgroups. The coefficients of the team quality variables in the countries outside the top five leagues show that: 1) the higher the away team quality, the higher the attendance, which is a confirmation that the quality of the opponents matters [31]; and 2) the lower the home team quality, the higher the attendance, potentially showing evidence of the 'David and Goliath effect'. Team quality is also an important factor for fans of clubs in the top five leagues, albeit with a lower coefficient. However, these fans are not significantly influenced by the quality of their own team, likely because it is on average higher than the opponents. This is consistent with previous studies in European football which show that opponents' quality matters to all fans, but more so to fans of lower ranked clubs [49,50,83].

Even though all the variables capturing the presence of star players show significant positive coefficients in the regressions conducted on the whole dataset, when analysing the two

**Table 3.  UCL attendance in the countries hosting the top five leagues.**

| | (1) | (2) | (3) | (4) | (5) | (6) | (7) |
|---|---|---|---|---|---|---|---|
| home_income | 0.008 | 0.010 | 0.008 | 0.009 | 0.010 | 0.010 | 0.010 |
| | (0.046) | (0.036) | (0.036) | (0.036) | (0.036) | (0.036) | (0.036) |
| away_income | 0.023** | 0.036*** | 0.037*** | 0.036*** | 0.036*** | 0.036*** | 0.035*** |
| | (0.012) | (0.013) | (0.013) | (0.013) | (0.013) | (0.013) | (0.013) |
| distance | -0.006 | -0.004 | -0.005 | -0.005 | -0.004 | -0.005 | -0.005 |
| | (0.009) | (0.010) | (0.010) | (0.010) | (0.010) | (0.010) | (0.010) |
| demperature | -0.0001 | 0.0002 | 0.0001 | 0.0001 | 0.0002 | 0.0002 | 0.0001 |
| | (0.001) | (0.001) | (0.001) | (0.001) | (0.001) | (0.001) | (0.001) |
| rain | 0.016 | 0.013 | 0.013 | 0.013 | 0.014 | 0.014 | 0.015 |
| | (0.021) | (0.022) | (0.022) | (0.022) | (0.022) | (0.022) | (0.022) |
| home_team_form_ | -0.002 | 0.026 | 0.028 | 0.028 | 0.024 | 0.023 | 0.025 |
| | (0.083) | (0.085) | (0.084) | (0.085) | (0.085) | (0.084) | (0.085) |
| away_team_form | 0.117 | 0.208** | 0.188** | 0.199** | 0.207** | 0.195** | 0.195** |
| | (0.081) | (0.082) | (0.083) | (0.082) | (0.082) | (0.082) | (0.083) |
| outcome_uncertainty | 0.120** | -0.049 | -0.038 | -0.039 | -0.050 | -0.044 | -0.040 |
| | (0.061) | (0.052) | (0.052) | (0.052) | (0.052) | (0.052) | (0.053) |
| group_stage | -0.076** | -0.104*** | -0.097*** | -0.101*** | -0.104*** | -0.097*** | -0.099*** |
| | (0.022) | (0.023) | (0.024) | (0.024) | (0.023) | (0.024) | (0.023) |
| last_16 | -0.013 | -0.015 | -0.010 | -0.014 | -0.015 | -0.010 | -0.012 |
| | (0.024) | (0.024) | (0.025) | (0.024) | (0.024) | (0.024) | (0.024) |
| quarter_final | -0.036 | -0.034 | -0.027 | -0.032 | -0.034 | -0.030 | -0.033 |
| | (0.024) | (0.024) | (0.025) | (0.024) | (0.024) | (0.024) | (0.024) |
| derby | 0.020 | 0.026 | 0.015 | 0.019 | 0.028 | 0.025 | 0.026 |
| | (0.037) | (0.038) | (0.038) | (0.038) | (0.038) | (0.038) | (0.038) |
| competitive_intensity_ko | -0.004 | -0.007 | -0.006 | -0.007 | -0.008 | -0.007 | -0.007 |
| | (0.008) | (0.008) | (0.008) | (0.008) | (0.008) | (0.008) | (0.008) |
| competitive_intensity_gs | -0.019 | -0.011 | -0.009 | -0.012 | -0.013 | -0.014 | -0.016 |
| | (0.032) | (0.030) | (0.030) | (0.030) | (0.030) | (0.030) | (0.030) |
| home_team_quality | 0.145 | | | | | | |
| | (0.116) | | | | | | |
| away_team_quality | 0.203*** | | | | | | |
| | (0.035) | | | | | | |
| bd_winner | | 0.013 | | | | | |
| | | (0.022) | | | | | |
| bd_3 | | | 0.031** | | | | |
| | | | (0.014) | | | | |
| bd_10 | | | | 0.009 | | | |
| | | | | (0.005) | | | |
| bd_winner_1 | | | | | 0.006 | | |
| | | | | | (0.015) | | |
| bd_3_1 | | | | | | 0.020 | |
| | | | | | | (0.012) | |
| bd_10_1 | | | | | | | 0.008 |
| | | | | | | | (0.006) |
| constant | 9.331*** | 9.507*** | 9.542*** | 9.533*** | 9.521*** | 9.518*** | 9.523*** |
| | (0.514) | (0.419) | (0.418) | (0.411) | (0.414) | (0.417) | (0.414) |
| Sigma | 0.028*** | 0.029*** | 0.029*** | 0.029*** | 0.029*** | 0.029*** | 0.029*** |

(*Continued*)

**Table 3.** (Continued)

| | (1) | (2) | (3) | (4) | (5) | (6) | (7) |
|---|---|---|---|---|---|---|---|
| | (0.003) | (0.003) | (0.003) | (0.003) | (0.003) | (0.003) | (0.003) |
| *Observations* | 746 | 746 | 746 | 746 | 746 | 746 | 746 |

*Notes*. Robust standard errors in parentheses obtained using the robust or sandwich estimator of variance: $p^* < 0.10$, $p^{**} < 0.05$, $p^{***} < 0.01$.

*Semi_final* omitted because of collinearity.

Season and team fixed effects omitted for sake of simplicity.

different subgroups the presence of star players is also much more attractive for fans of clubs outside the top five leagues. Indeed, the star player variable is only significant in one specification for fans of the top five leagues, whereas it is significant in all six specifications for fans of clubs in the other leagues. This suggests that the presence of star players is more of a motivating factor for fans who are less accustomed to the presence of such players, and for whom the UCL is the only competition that offers the potential to watch such star players.

## Competition structure (stage of competition and derby)

The analysis conducted on the whole dataset shows that group stage matches are less attractive. However, there is no evidence that the stage of the competition significantly affects attendance levels for fans of clubs outside of the top five leagues, although the strength of the coefficient does increase during the latter stages. Our findings do suggest that group stage matches are less attractive only for fans of clubs in the top five leagues. This finding, along with the fans' preference for better opponents, may support one of the arguments of the ESL promoters that the current format is not particularly attractive as it allows only a limited number of matches between elite clubs before the knockout stage. This is due to the seeding format of the group stage, where the UCL title holder, Europa League title holder and league champions of the top six ranked associations are placed into 'Pot 1' and cannot be drawn against each other. However, it should be recognised that this finding is only within the *existing* context of the competition, i.e., where there is a certain level of outcome uncertainty and where elite clubs are not guaranteed entry into the competition every season, as was planned in the recent ESL proposals.

Derby matches involving two teams from the same country also do not appear to attract a higher crowd due to a more intense rivalry. This runs counter to previous findings in domestic league competitions in European football [37,84,85], but is consistent with Valenti et al's [51] findings in the women's UCL. It should be noted that derby matches are not widespread in the UCL, as clubs from the same country cannot be drawn into the same group and are also kept apart in the first knockout round draw, meaning that there were only 15 instances of derby matches in our entire dataset.

## Explanatory variables (income, distance, temperature and rain)

The regional income of away teams is a significant predictor of attendance levels for matches in all but one specification, although it should be noted that the strength of the coefficient is also relatively small throughout. This suggests that the purchasing power of away team fans affects their ability to travel and support their team in international matches. Although interestingly distance, a proxy for the travelling costs for away supporters, is not a significant factor in any specification. This may be because most matches require significant international travel, with the average distance to a match being 1,541 kilometres. As a result, the opportunity cost of attending an away match is similar regardless of which area of Europe you are travelling

**Table 4. UCL attendance in the other countries.**

|  | (1) | (2) | (3) | (4) | (5) | (6) | (7) |
|---|---|---|---|---|---|---|---|
| home_income | 0.600*** | 0.527*** | 0.508*** | 0.527*** | 0.544*** | 0.542*** | 0.524*** |
|  | (0.066) | (0.056) | (0.060) | (0.060) | (0.056) | (0.056) | (0.059) |
| away_income | 0.018 | 0.058*** | 0.058*** | 0.051*** | 0.058*** | 0.058*** | 0.052*** |
|  | (0.017) | (0.017) | (0.017) | (0.017) | (0.017) | (0.017) | (0.017) |
| distance | 0.000 | -0.008 | -0.007 | -0.008 | -0.001 | -0.006 | -0.008 |
|  | (0.021) | (0.022) | (0.021) | (0.021) | (0.022) | (0.021) | (0.021) |
| temperature | 0.007*** | 0.006*** | 0.006*** | 0.007*** | 0.006*** | 0.006*** | 0.006*** |
|  | (0.002) | (0.002) | (0.002) | (0.002) | (0.002) | (0.002) | (0.002) |
| rain | -0.001 | -0.003 | -0.002 | 0.002 | -0.001 | 0.003 | 0.006 |
|  | (0.031) | (0.031) | (0.031) | (0.031) | (0.031) | (0.031) | (0.031) |
| home_team_form | 0.217 | 0.319* | 0.305* | 0.285* | 0.311* | 0.335** | 0.338** |
|  | (0.189) | (0.194) | (0.193) | (0.193) | (0.194) | (0.193) | (0.196) |
| away_team_form | -0.256* | -0.038 | -0.080 | -0.137 | -0.049 | -0.090 | -0.139 |
|  | (0.157) | (0.151) | (0.151) | (0.154) | (0.154) | (0.150) | (0.153) |
| outcome_uncertainty | 0.016 | 0.001 | -0.026 | -0.057 | 0.009 | -0.037 | -0.061 |
|  | (0.080) | (0.082) | (0.081) | (0.080) | (0.084) | (0.082) | (0.079) |
| group_stage | 0.093 | 0.115 | 0.118 | 0.173 | 0.099 | 0.118 | 0.194 |
|  | (0.087) | (0.092) | (0.091) | (0.092) | (0.091) | (0.089) | (0.093) |
| last_16 | 0.233 | 0.261 | 0.268 | 0.323 | 0.246 | 0.273 | 0.334 |
|  | (0.086) | (0.094) | (0.093) | (0.094) | (0.093) | (0.091) | (0.097) |
| quarter_final | 0.323 | 0.408 | 0.359 | 0.395 | 0.394 | 0.377 | 0.375 |
|  | (0.117) | (0.132) | (0.141) | (0.129) | (0.131) | (0.124) | (0.125) |
| competitive_intensity_ko | -0.018 | -0.017 | -0.019 | -0.018 | -0.016 | -0.022 | -0.009 |
|  | (0.026) | (0.026) | (0.028) | (0.026) | (0.027) | (0.024) | (0.026) |
| competitive_intensity_gs | -0.133* | -0.130* | -0.142* | -0.125 | -0.127 | -0.146* | -0.142* |
|  | (0.107) | (0.102) | (0.101) | (0.101) | (0.103) | (0.099) | (0.102) |
| home_team_quality | -0.962*** |  |  |  |  |  |  |
|  | (0.402) |  |  |  |  |  |  |
| away_team_quality | 0.341*** |  |  |  |  |  |  |
|  | (0.062) |  |  |  |  |  |  |
| bd_winner |  | 0.181*** |  |  |  |  |  |
|  |  | (0.074) |  |  |  |  |  |
| bd_3 |  |  | 0.102*** |  |  |  |  |
|  |  |  | (0.034) |  |  |  |  |
| bd_10 |  |  |  | 0.064*** |  |  |  |
|  |  |  |  | (0.018) |  |  |  |
| bd_winner_1 |  |  |  |  | 0.179*** |  |  |
|  |  |  |  |  | (0.061) |  |  |
| bd_3_1 |  |  |  |  |  | 0.125*** |  |
|  |  |  |  |  |  | (0.026) |  |
| bd_10_1 |  |  |  |  |  |  | 0.069*** |
|  |  |  |  |  |  |  | (0.016) |
| _cons | 4.342*** | 4.029*** | 4.253*** | 4.140*** | 3.817*** | 3.895*** | 4.132*** |
|  | (0.758) | (0.765) | (0.795) | (0.789) | (0.758) | (0.750) | (0.774) |
| Sigma | 0.058*** | 0.062*** | 0.062*** | 0.061*** | 0.062*** | 0.061*** | 0.060*** |
|  | (0.007) | (0.007) | (0.007) | (0.007) | (0.007) | (0.007) | (0.007) |

(*Continued*)

**Table 4.**  (Continued)

|  | (1) | (2) | (3) | (4) | (5) | (6) | (7) |
|---|---|---|---|---|---|---|---|
| *Observations* | 488 | 488 | 488 | 488 | 488 | 488 | 488 |

*Notes*. Robust standard errors in parentheses obtained using the robust or sandwich estimator of variance: p*<0.10, p**<0.05, p***<0.01.

*Semi_final* omitted because of collinearity.

*derby* omitted because no game between clubs from the same country were played in this sub-group.

Season and team fixed effects omitted for sake of simplicity.

from and to, meaning that increased travel time does not deter fans from attending UCL matches in the same way it does for domestic leagues [33,83].

However, the importance of the regional income levels of home fans–whose coefficients are significant and positive when analysing the whole dataset–is not consistent between leagues. The purchasing power of home team fans is not significant for clubs in the top five leagues, but is a highly significant predictor of attendance for those clubs outside the top five leagues. This may reflect the different income levels in these regions, with an average income per capita in the top five leagues of €50,212, compared to €27,675 in the other leagues. Whereas no region in the top five leagues has an average income per capita below €19,000, nearly half of the regions in the other leagues have an average income per capita below this level.

Finally, favourable weather conditions are not strongly associated with higher attendances in the UCL, with the presence of rain insignificant for clubs in all leagues and higher temperatures only significant for clubs outside the top five leagues, but with a very small coefficient. The significance of temperature in the other leagues may be due to the temperature of matches played in these countries were, on average, 2 Celsius lower. These matches also experienced more extreme cold weather, with temperatures as low as -9 Celsius during some games compared to -3 Celsius at the lowest in the top five leagues.

## Discussion

In this section, we critically reflect on the findings of our empirical analysis and consider the implications for UEFA, European football clubs and their fans. In doing so, it is important to also recognise the pressure that continues to be applied to UEFA by elite European clubs through the threat of a breakaway European Super League (ESL). As noted earlier in the paper, the greatest challenge to the UCL came on 18 April 2021, when 12 elite clubs from England, Italy and Spain announced they would be leaving the UCL to form a separate ESL. However, following fierce opposition from their own fans, as well as from UEFA and FIFA (the world governing body) and some national governments, participating clubs began to withdraw from the proposed competition and operations were suspended just three days after being officially announced [86]. Despite this, it should be noted that the announcement of the ESL coincided with format changes to the UCL, which are due to be instated from the 2024/25 season [87]. Under these changes the tournament will expand the number of clubs competing to 36 and introduce a 'Swiss model' format for the opening round. Rather than the current opening round of eight groups, there will now be a single league of all 36 clubs and each team will play fixtures against ten different clubs. Preferential qualification has also been offered to elite clubs, with two of the additional qualification spots awarded to the clubs with the highest UEFA club coefficients that have not already qualified for the UCL. The ongoing and highly charged debate around the threat posed by an ESL and the future of the UCL [88] adds an extra dimension to this discussion, and this paper's analysis enables for an evidence-based discussion surrounding the restructured format of the UCL.

Before proceeding to the discussion, however, it is important to set our findings in the wider context of sport consumption. We have focused here on the determinants of stadium attendance, making a contribution to a very well-established academic literature. However, it is the case that, for many clubs now, broadcasting revenue (largely from people watching on television) outstrips stadium-based revenue (i.e., tickets and associated purchases). In this sense, we do not claim that our findings should be generalised to all spectators. Having said this, recent analysis of the factors that appear to influence television audiences for the men's UEFA Champions League found similar results to our analysis here. Indeed, Wills et al. [13] found that the presence of star players and higher team quality had a significant, positive effect on TV audiences, while uncertainty of outcome was typically not significant. As such, while it still important not to over-claim, we can be more confident that our conclusions are not limited to match-going fans.

Our analysis gives a good indication that the quality of the visiting team is an important motivating factor for match-going fans, and especially for those fans outside the top five leagues, for whom the presence of star players is also a significant factor. When the plans for a breakaway European Super League were announced in April 2021, it was proposed that the league would have 15 permanent 'founding clubs'. Twelve of these clubs announced their intention to join–Arsenal, Chelsea, Liverpool, Manchester United, Manchester City, Tottenham Hotspur, Juventus, Inter Milan, AC Milan, Real Madrid, Atlético Madrid and Barcelona–and it was suggested that the final three slots were intended for Bayern Munich, Borussia Dortmund and Paris Saint-Germain. As Table 5 shows, over the ten seasons included in our

**Table 5. Top 10 squad values and Ballon d'Or players by season.**

| Season | Top 10 squad values | Founding club % | Top 10 Ballon d'Or players | Founding club % |
|---|---|---|---|---|
| 2009/10 | Real Madrid, Chelsea, Barcelona, Inter Milan, Manchester United, Liverpool, AC Milan, Juventus, Bayern Munich, Arsenal | 100% | Messi (Barcelona), Iniesta (Barcelona), Xavi (Barcelona), Sneijder (Inter), Forlan (Atletico), Ronaldo (Real), Casillas (Real), Villa (Barcelona), Drogba (Chelsea), Xabi Alonso (Real) | 100% |
| 2010/11 | Barcelona, Real Madrid, Chelsea, Inter Milan, Manchester United, AC Milan, Arsenal, Tottenham, Bayern Munich, Roma | 90% | Messi (Barcelona), Ronaldo (Real), Xavi (Barcelona), Iniesta (Barcelona), Rooney (Man United), Suarez (Liverpool), Forlan (Inter), Eto'o (Inter), Casillas (Real), Neymar (Santos) | 90% |
| 2011/12 | Barcelona, Real Madrid, Chelsea, Manchester City, Manchester United, Inter Milan, Bayern Munich, Arsenal, AC Milan, Porto | 90% | Messi (Barcelona), Ronaldo (Real), Iniesta (Barcelona), Xavi (Barcelona), Falcao (Atletico), Casillas (Real), Pirlo (Juventus), Drogba (Chelsea), van Persie (Arsenal) Ibrahimovic (AC Milan) | 100% |
| 2012/13 | Barcelona, Real Madrid, Manchester City, Manchester United, Chelsea, Bayern Munich, PSG, Juventus, Arsenal, AC Milan | 100% | Ronaldo (Real Madrid), Messi (Barcelona), Ribery (Bayern Munich), Ibrahimovic (PSG), Neymar (Santos), Iniesta (Barcelona), van Persie (Man United), Robben (Bayern), Bale (Tottenham), Pirlo (Juventus) | 90% |
| 2013/14 | Barcelona, Real Madrid, Chelsea, Man United, Bayern Munich, Man City, Juventus, PSG, Arsenal, AC Milan | 100% | Ronaldo (Real), Messi (Barcelona), Neuer (Bayern), Robben (Bayern), Muller (Bayern), Lahm (Bayern), Neymar (Barcelona), Rodriguez (Monaco), Kroos (Bayern), Di Maria (Real) | 90% |
| 2014/15 | Real Madrid, Barcelona, Bayern Munich, Chelsea, Man City, Arsenal, PSG, Juventus, Borussia Dortmund, Atletico Madrid | 100% | Messi (Barcelona), Ronaldo (Real), Neymar (Barcelona), Lewandowski (Bayern), Suarez (Barcelona), Muller (Bayern), Meuer (Bayern), Hazard (Chelsea), Iniesta (Barcelona), Sanchez (Arsenal) | 100% |
| 2015/16 | Real Madrid, Barcelona, Bayern Munich, Chelsea, Manchester City, PSG, Arsenal, Manchester United, Atletico Madrid, Juventus | 100% | Ronaldo (Real), Messi (Barcelona), Griezmann (Atletico), Suarez (Barcelona), Neymar (Barcelona), Bale (Real), Mahrez (Leicester), Vardy (Leicester), Buffon (Juventus), Pepe (Real) | 80% |
| 2016/17 | Barcelona, Real Madrid, Manchester City, Bayern Munich, Arsenal, PSG, Atletico Madrid, Juventus, Tottenham, Borussia Dortmund | 100% | Ronaldo (Real), Messi (Barcelona), Neymar (Barcelona), Buffon (Juventus), Modric (Real), Ramos (Real), Mbappe (Monaco), Kante (Chelsea), Lewandowski (Bayern), Kane (Tottenham) | 90% |
| 2017/18 | Barcelona, Real Madrid, Chelsea, Manchester United, Manchester City, Bayern Munich, Atletico Madrid, PSG, Juventus, Tottenham | 100% | Modric (Real), Ronaldo (Real), Griezmann (Atletico), Mbappe (PSG), Messi (Barcelona), Salah (Liverpool), Varane (Real), Hazard (Chelsea), De Bruyne (Man City), Kane (Tottenham) | 100% |
| 2018/19 | Barcelona, Real Madrid, Manchester City, Liverpool, Juventus, Atletico Madrid, PSG, Bayern Munich, Manchester United, Tottenham | 100% | Messi (Barcelona), van Dijk (Liverpool), Ronaldo (Juventus), Mane (Liverpool), Mbappe (PSG), Alisson (Liverpool), Lewandowski (Bayern), Silva (Man City), Mahrez (Man City) | 100% |
| | Average | 98% | Average | 94% |

analysis, these 15 'founding clubs' made up the top ten most valuable rosters (our measure of team quality) 98 percent of the time and their squads contained the top ten Ballon d'Or players (our measure of star players) 94 percent of the time.

This underlines how important it is for UEFA to ensure these elite clubs remain involved in the UCL to maximise the levels of team quality and to guarantee the involvement of star players. One way in which UEFA can encourage the continued involvement of elite clubs is to ensure the competition format remains relevant to these club's fans. In our analysis of the top five leagues' stadium attendance (Table 3), 512 of the 746 matches (69 percent) involved one or more of the top 15 elite clubs. As a result, this analysis can be taken as a good proxy for the demands of elite clubs' fans. The indication from our analysis is that elite clubs' match-going fans are primarily interested in watching matches with high quality opposition and that attendances from group-stage matches are currently suffering, perhaps due to this lack of quality. UEFA's new 'Swiss model' is a noticeable attempt to remedy this situation, by moving from the current format, in which elite clubs are kept apart in the group stage through seeding, to a format in which elite clubs will be guaranteed at least two matches against other top ranked 'Pot 1' clubs in the group stage. UEFA themselves claim this will bring 'more opportunities to see Europe's top teams playing each other earlier in the competition' [87] and our analysis suggests this will be well-received by elite clubs' fans and lead to higher interest in the group stages.

It is also important that UEFA meets the needs of fans and clubs from outside the top five leagues and supports the wider European football pyramid. Previous empirical analysis has shown that star players can have positive externalities on the other clubs they play against and leagues they compete in, generating income for more than just their employer [6,8,66,89,90]. This gives competition organisers the potential to leverage the value of star players and the elite clubs they represent. UEFA's new 'Swiss model' format for the UCL seeks to take advantage of this potential, with each club now meeting ten separate clubs in the opening stage, rather than the same three clubs twice under the previous format. Consequently, those elite clubs containing the Ballon d'Or winner, or with the highest overall team quality, will meet a greater number of clubs in the group stage every year. This gives smaller clubs more opportunities to host elite clubs and generate additional income from these matches. In light of this, UEFA's plan to offer UCL qualification to clubs with the highest club coefficient, but which do not qualify through their domestic league position, should not be automatically dismissed as elitist. These policies will also have the effect of protecting the value of the competition for all clubs by ensuring the presence of star players and elite clubs, in turn maximising interest in the competition.

While recognising that UEFA must consider broader issues than stadium attendance alone, the importance of maximising match day revenue for European football clubs should not be underestimated. The recent financial struggles felt by clubs during the COVID-19 pandemic has highlighted this, with the restrictions on crowds during the 2019/20 and 2020/21 seasons causing an estimated revenue loss of €4.4bn across European leagues [91]. As noted earlier, Wills et al.'s [13] examination of TV viewing figures in the men's UCL found similar evidence for fan preferences, with both star players and higher team quality having a significant positive effect on TV audiences. Given broadcasting revenues represent the largest source of income for the competition, our analysis suggests that the new "Swiss model' structure for the UCL represents a positive evolution from the previous format. This new format gives greater opportunities for clubs both within and outside the top five leagues to host elite clubs and star players. It is this unique characteristic that sets the UCL apart from domestic leagues, and our analysis suggests UEFA should continue to leverage this in order to meet fans' demands.

## Limitations and future research

As with any study, the analysis undertaken here has some limitations, due to the unavailability of certain data. First, it was not possible to retrieve data on ticket prices, which limits the understanding of the specific price effect on attendance. Second, it was not possible to differentiate between the number of home and away fans present at each match, meaning we were unable to account for behavioural differences in certain matches where there were a greater number of away fans. Third, it was not possible to differentiate between the proportion of season tickets and match-day tickets sold, which has the potential to distort findings where there could have been greater fluctuations in attendance levels had the tickets not already been allocated to season ticket holders. However, despite these limitations, which affect most stadium attendance demand analysis, we believe the findings are robust enough to draw reliable conclusions on UCL fan behaviour.

Nevertheless, there is a great deal more that researchers in this area could do. As Schreyer and Ansari [4] urged, we need more research on stadium attendance demand in international cup competitions. This study is one major step, but, as just one example, the Europa League, UEFA's second-tier club competition, has been running for 50 years and is also lacking any empirical analysis of stadium attendance demand. A greater understanding of this competition would offer an important insight for UEFA, who have expanded their competition offering further in the 2021/22 season, with the launch of a third-tier club competition, the UEFA Europa Conference League. Similarly, Schreyer and Ansari's [4] scoping review also noted a lack of focus on domestic cup competitions and women's sports, analysis of which would allow for a greater comparison between the determinants of demand for different competition formats.

## Conclusion

This paper has provided the first empirical analysis of stadium attendance demand for the men's UEFA Champions League. It has found that match-going fans in all leagues are not primarily interested in uncertainty of outcome or competitive intensity, at least within the current context that has a certain degree of outcome uncertainty built in. Instead, the quality of the away team appears to be the one factor that is important to all fans. This factor is especially important to fans outside the top five leagues, who are also motivated by the presence of star players in matches, indicating that these fans value the UCL for its unique ability to bring the world's best clubs and players to their club's stadium. It is this characteristic that sets the UCL apart from domestic leagues, where factors such as competitive intensity are more important, and it is therefore important that UEFA ensures the continued involvement of elite clubs in the UCL, while balancing the needs of smaller clubs and the wider football pyramid. Our analysis suggests that the UCL's most recent evolution towards a 'Swiss model' tournament structure will better meet these fan demands, while continuing to balance the needs of elite clubs and the wider football pyramid.

## Supporting information

**S1 File. Stata test and Tobit regressions with 100% stadium capacity cut-off points.** This log file shows all the regressions and tests conducted on Stata by using a Tobit model with 100% stadium capacity cut-off points.
(LOG)

**S2 File. Stata test and Tobit regressions with 90% stadium capacity cut-off points.** This log file shows all the regressions and tests conducted on Stata by using a Tobit model with 90%

stadium capacity cut-off points.
(LOG)

**S3 File. Stata test and Tobit regressions with 95% stadium capacity cut-off points.** This log file shows all the regressions and tests conducted on Stata by using a Tobit model with 95% stadium capacity cut-off points.
(LOG)

## Author Contributions

**Conceptualization:** George Wills, Richard Tacon.

**Data curation:** George Wills.

**Formal analysis:** George Wills.

**Methodology:** Francesco Addesa.

**Supervision:** George Wills.

**Writing – original draft:** George Wills, Francesco Addesa.

**Writing – review & editing:** George Wills, Richard Tacon.

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
