## [Decision Letter · Decision Letter 0]

18 Jul 2022

PONE-D-22-06932Stadium attendance demand in the men’s UEFA Champions League: Do fans value sporting contest or match quality?PLOS ONE

Dear Dr. Addesa,

Thank you for submitting your manuscript to PLOS ONE. After careful consideration, we feel that it has merit but does not fully meet PLOS ONE’s publication criteria as it currently stands. Therefore, we invite you to submit a revised version of the manuscript that addresses the points raised during the review process.

Your manuscript has been assessed by two expert reviewers, whose comments are appended below. The reviewers have highlighted concerns about the formulation of the statistical model and the framing of some of the conclusions. Please ensure you respond to each point carefully in your response to reviewers document, and modify your manuscript accordingly.

We look forward to receiving your revised manuscript.

Kind regards,

Joseph Donlan

Editorial Office

PLOS ONE

Journal Requirements:

Reviewers' comments:

Reviewer's Responses to Questions

**Comments to the Author**

1. Is the manuscript technically sound, and do the data support the conclusions?

Reviewer #1: Partly

Reviewer #2: Yes

2. Has the statistical analysis been performed appropriately and rigorously? 

Reviewer #1: Yes

Reviewer #2: Yes

3. Have the authors made all data underlying the findings in their manuscript fully available?

Reviewer #1: No

Reviewer #2: Yes

4. Is the manuscript presented in an intelligible fashion and written in standard English?

Reviewer #1: Yes

Reviewer #2: Yes

5. Review Comments to the Author

Reviewer #1: An interesting analysis, with some novelty (the dataset; inclusion of notions of competitive intensity an indicators for presence of star players). However, I think the conclusions are overstated. In my view the analysis tells us little about the determinants for the consumption of sport. This is because: 1) I think the model is too crude, in particular the response variable, because attendance is most closely linked to stadium capacity and I couldn't see how this was accommodated in the model; 2) spectators of a contest form only a very small fraction of consumers of the contest as most watch on television; 3) high quality opposition and uncertainty of outcome must surely be co-linear, so it is not exactly clear what your analysis has established. I would therefore suggest a revision to include a more critical analysis of your own findings.

Reviewer #2: [Preliminary remarks] Even though I am fully aware of the fact that reviewers must/should not assess PLOS ONE submissions regarding their potential relevance and impact, I would like to stress that I think that this submission is highly relevant. More specifically, as the authors note, research on stadium attendance demand has refrained from analyzing cup competitions. However, understanding what nurtures spectators’ interest in such sporting events is undoubtedly necessary. This is particularly true for an event as popular (and prestigious) as the men’s UEFA Champion’s League. I must admit, though, that, even if I applaud the authors on approaching this overlooked empirical setting, I am getting increasingly tired of studies focusing on match outcome uncertainty as an antecedent of stadium attendance demand.

Below, I will address whether the manuscript fulfills PLOS ONE's seven publication criteria. In contrast, unlike in most other cases, I refrain from assessing how the study is motivated and how well and nuanced the authors interact with the literature.

[01] The study presents the results of original research.

Yes, definitely. The authors communicate original empirical research.

[02] Results reported have not been published elsewhere.

Yes. To my knowledge, this is the first manuscript modeling spectator interest in the men’s UEFA Champions League. During brief desk research, I was not able to find it already online.

[03] Experiments, statistics, and other analyses are performed to a high technical standard and are described in sufficient detail.

I mostly agree. In line with most stadium attendance research of higher quality, the authors employ a Tobit model using a logarithmized dependent variable. However, in terms of a sufficient description of the methodological set-up, I would like to understand better (technically) why the authors logarithmized/transformed the dependent variable and why the cut-off points are stadium capacity rather than, as it is more commonly in the literature, 90 or 95 percent of it (or whether the chosen level affects the results). Further, I think the authors should discuss the nature of their dependent variable in a bit more detail, incl. the potential limitations arising from exploiting such public data (e.g., how exactly is spectator demand affected by distributing free and season tickets and the resulting no-show behavior, most notably during group stages). Finally, to better assess the robustness of the results, I think it would be helpful also to see a model across all matches rather than only presenting the two subsets.

[04] Conclusions are presented in an appropriate fashion and are supported by the data.

I agree.

[05] The article is presented in an intelligible fashion and is written in standard English.

I agree, even though I must admit that I think the contribution-to-length ratio is not ideal, and the article could have been much shorter. However, as page limits are not a factor, this is, perhaps, neglectable.

[06] The research meets all applicable standards for the ethics of experimentation and research integrity.

I agree.

[07] The article adheres to appropriate reporting guidelines and community standards for data availability.

I agree.

6. PLOS authors have the option to publish the peer review history of their article (what does this mean?). If published, this will include your full peer review and any attached files.

Reviewer #1: No

Reviewer #2: No

---

## [Author Response · Author response to Decision Letter 0]

6 Sep 2022

Thank you for submitting your manuscript to PLOS ONE. After careful consideration, we feel that it has merit but does not fully meet PLOS ONE’s publication criteria as it currently stands. Therefore, we invite you to submit a revised version of the manuscript that addresses the points raised during the review process.

Your manuscript has been assessed by two expert reviewers, whose comments are appended below. The reviewers have highlighted concerns about the formulation of the statistical model and the framing of some of the conclusions. Please ensure you respond to each point carefully in your response to reviewers document, and modify your manuscript accordingly.

Authors: Thank you for the invitation to resubmit. Please see below our specific responses about the statistical model and the framing of our conclusions.

Reviewer 1

An interesting analysis, with some novelty (the dataset; inclusion of notions of competitive intensity an indicators for presence of star players). However, I think the conclusions are overstated. In my view the analysis tells us little about the determinants for the consumption of sport. This is because…

Authors: We thank Reviewer 1 for their valuable feedback, in particular about ensuring our conclusions are proportionate to the analysis we have conducted. Please see below for our specific responses to the points you raised.

1) I think the model is too crude, in particular the response variable, because attendance is most closely linked to stadium capacity and I couldn't see how this was accommodated in the model;

Authors: In designing our model, we have followed the procedures that are typically used in academic analysis of stadium demand, in order to account for stadium capacity. Specifically, we used a Tobit model, which is designed to account for truncation due to stadium capacity, and we added home team fixed effects, which capture the home team’s stadium capacity. This takes into account that bigger, more popular and richer clubs have bigger stadiums (see Besters LM., van Ours JC, van Tuijl MA. How outcome uncertainty, loss aversion and team quality affect stadium attendance in Dutch professional football. J Econ Psychol. 2019;72: 117-127). 

However, Reviewer 2 also raised an issue about our dependent variable, which we have addressed through additional analysis and supplying the results. We hope that our response to this also responds to your comment here. Please see our comment below on this and the changes in the manuscript.

2) spectators of a contest form only a very small fraction of consumers of the contest as most watch on television;

Authors: This is undeniably true. Our empirical analysis focuses on stadium attendance and, in this sense, contributes to the well-established literature on stadium attendance demand. However, you are of course right that this sits within a sport consumption context where, for many clubs, broadcasting revenue far outstrips match-day revenue.

We felt that we had set this in context and did not try to over-claim. However, having read your comment, we reread the paper and especially the discussion and conclusions and felt that, as you point out, we were not as clear on this as we could have been.

We have addressed this by adding a specific paragraph just ahead of our main discussion and also by trying to temper the language of our conclusions – specifying ‘match-going’ fans, where relevant, and modifying our language somewhat to attempt a more balanced-sounding set of conclusions.

Having said this, the fact that recent analysis of TV audiences of the men’s UCL found similar things to our analysis of stadium attendance does suggest that the preferences of match-going fans for this competition are not wildly ‘out of sync’ with their TV-watching counterparts.

3) high quality opposition and uncertainty of outcome must surely be co-linear, so it is not exactly clear what your analysis has established.

Authors: This is not the case, as the correlation between quality of opposition and outcome uncertainty is not necessarily significant. For a ‘big’ club, i.e., a high-quality team, it is true that the higher the opposition quality, ceteris paribus, the higher the uncertainty of outcome. However, for a ‘small’ club, the higher the opposition quality, the lower the uncertainty of outcome, in that it is much more likely they will lose the match. This is borne out by the data. We have provided evidence in our log file that the correlation between the two variables in our sample is moderately weak.

I would therefore suggest a revision to include a more critical analysis of your own findings.

Authors: We hope that the responses to the points above, plus the response to Reviewer 2’s comments does this.

Reviewer 2

Even though I am fully aware of the fact that reviewers must/should not assess PLOS ONE submissions regarding their potential relevance and impact, I would like to stress that I think that this submission is highly relevant. More specifically, as the authors note, research on stadium attendance demand has refrained from analyzing cup competitions. However, understanding what nurtures spectators’ interest in such sporting events is undoubtedly necessary. This is particularly true for an event as popular (and prestigious) as the men’s UEFA Champion’s League. I must admit, though, that, even if I applaud the authors on approaching this overlooked empirical setting, I am getting increasingly tired of studies focusing on match outcome uncertainty as an antecedent of stadium attendance demand.

Authors: Thank you for your comments on the relevance of the article. As for your final comment, we do recognise this and do recognise that there are certainly other avenues to explore! Still, we felt, as you acknowledged, that analysis of this particular competition, in relation to match outcome uncertainty (and other factors), was warranted. 

[03] Experiments, statistics, and other analyses are performed to a high technical standard and are described in sufficient detail.

I mostly agree. In line with most stadium attendance research of higher quality, the authors employ a Tobit model using a logarithmized dependent variable. However, in terms of a sufficient description of the methodological set-up, I would like to understand better (technically) why the authors logarithmized/transformed the dependent variable and why the cut-off points are stadium capacity rather than, as it is more commonly in the literature, 90 or 95 percent of it (or whether the chosen level affects the results).

Authors: We note your point here and thank you for raising it.

The use of log variables aims at interpreting the estimated coefficients as elasticities. It is true that the use of 90 or 95 percent cut-off points is more common. However, there are still a number of studies (e.g., Meehan Jr, J. W., Nelson, R. A., & Richardson, T. V. (2007). Competitive balance and game attendance in Major League Baseball. Journal of Sports Economics, 8(6), 563-580; Hong, S., Mondello, M., & Coates, D. (2013). An examination of the effects of the recent economic crisis on major league baseball attendance demand. International Journal of Sport Finance, 8(2), 140-156) that conduct censored regressions using 100% capacity as cut-off points. We reckon that in the vast majority of our observations the security reasons that may reduce the actual stadium capacity do not occur. However, we have also run the regressions using 90 or 95 percent cut-off points (log files attached): the results do not significantly change.

[03 continued]

Further, I think the authors should discuss the nature of their dependent variable in a bit more detail, incl. the potential limitations arising from exploiting such public data (e.g., how exactly is spectator demand affected by distributing free and season tickets and the resulting no-show behavior, most notably during group stages).

Authors: We have now discussed the nature of the dependent variable in more detail and its potential limitations, linked to the free tickets, spectator no-show behaviour and the missing differentiation between season and match-day tickets.

[03 continued]

Finally, to better assess the robustness of the results, I think it would be helpful also to see a model across all matches rather than only presenting the two subsets.

Authors: We have now added the results for the whole dataset as well.

---

## [Decision Letter · Decision Letter 1]

6 Oct 2022

Stadium attendance demand in the men’s UEFA Champions League: Do fans value sporting contest or match quality?

PONE-D-22-06932R1

Dear Dr. Addesa,

We’re pleased to inform you that your manuscript has been judged scientifically suitable for publication and will be formally accepted for publication once it meets all outstanding technical requirements.

Kind regards,

Rafael Franco Soares Oliveira

Academic Editor

PLOS ONE

Additional Editor Comments (optional):

Dear authors,

Congratulations on your work! My recommendation is to accept.

Best regards

Reviewers' comments:

Reviewer's Responses to Questions

**Comments to the Author**

1. If the authors have adequately addressed your comments raised in a previous round of review and you feel that this manuscript is now acceptable for publication, you may indicate that here to bypass the “Comments to the Author” section, enter your conflict of interest statement in the “Confidential to Editor” section, and submit your "Accept" recommendation.

Reviewer #1: (No Response)

Reviewer #2: All comments have been addressed

2. Is the manuscript technically sound, and do the data support the conclusions?

Reviewer #1: (No Response)

Reviewer #2: (No Response)

3. Has the statistical analysis been performed appropriately and rigorously? 

Reviewer #1: (No Response)

Reviewer #2: (No Response)

4. Have the authors made all data underlying the findings in their manuscript fully available?

Reviewer #1: (No Response)

Reviewer #2: (No Response)

5. Is the manuscript presented in an intelligible fashion and written in standard English?

Reviewer #1: (No Response)

Reviewer #2: (No Response)

6. Review Comments to the Author

Reviewer #1: I think your responses to my comments are satisfactory. However, I would have liked to have seen exactly what changes you have made to the manuscript. I think new material (text, tables, figures) should be indicated e.g. new text should be shown red and deleted text crossed out.

Reviewer #2: Dear authors, thanks for resubnitting your manuscrit. I see my few remaining points addressed in this revised version.

7. PLOS authors have the option to publish the peer review history of their article (what does this mean?). If published, this will include your full peer review and any attached files.

Reviewer #1: No

Reviewer #2: No

---

## [Editor Report · Acceptance letter]

10 Oct 2022

PONE-D-22-06932R1 

Stadium attendance demand in the men’s UEFA Champions League: Do fans value sporting contest or match quality? 

Dear Dr. Addesa:

I'm pleased to inform you that your manuscript has been deemed suitable for publication in PLOS ONE. Congratulations! Your manuscript is now with our production department. 

Kind regards, 

on behalf of

Dr. Rafael Franco Soares Oliveira 

Academic Editor

PLOS ONE